# The Bee Gut Microbiota: Bridging Infective Agents Potential in the One Health Context

**DOI:** 10.3390/ijms25073739

**Published:** 2024-03-27

**Authors:** Bruno Tilocca, Viviana Greco, Cristian Piras, Carlotta Ceniti, Mariachiara Paonessa, Vincenzo Musella, Roberto Bava, Ernesto Palma, Valeria Maria Morittu, Anna Antonella Spina, Fabio Castagna, Andrea Urbani, Domenico Britti, Paola Roncada

**Affiliations:** 1Department of Health Science, University “Magna Graecia” of Catanzaro, 88100 Catanzaro, Italy; c.piras@unicz.it (C.P.); ceniti@unicz.it (C.C.); mariachiara.paonessa001@studenti.unicz.it (M.P.); musella@unicz.it (V.M.); roberto.bava@unicz.it (R.B.); palma@unicz.it (E.P.); morittu@unicz.it (V.M.M.); aa.spina@unicz.it (A.A.S.); castagnafabio@yahoo.it (F.C.); britti@unicz.it (D.B.); 2Department of Basic Biotechnological Sciences, Intensivological and Perioperative Clinics, Catholic University of the Sacred Hearth, 00168 Rome, Italy; viviana.greco@unicatt.it (V.G.); andrea.urbani@unicatt.it (A.U.); 3Unity of Chemistry, Biochemistry and Clinical Molecular Biology, Department of Diagnostic and Laboratory Medicine, Fondazione Policlinico Universitario A. Gemelli IRCCS, 00168 Rome, Italy

**Keywords:** bee functional microbiota, metaproteomics, One Health, bee gut bacteria, bee gut fungi, bee gut parasites

## Abstract

The bee gut microbiota plays an important role in the services the bees pay to the environment, humans and animals. Alongside, gut-associated microorganisms are vehiculated between apparently remote habitats, promoting microbial heterogeneity of the visited microcosms and the transfer of the microbial genetic elements. To date, no metaproteomics studies dealing with the functional bee microbiota are available. Here, we employ a metaproteomics approach to explore a fraction of the bacterial, fungal, and unicellular parasites inhabiting the bee gut. The bacterial community portrays a dynamic composition, accounting for specimens of human and animal concern. Their functional features highlight the vehiculation of virulence and antimicrobial resistance traits. The fungal and unicellular parasite fractions include environment- and animal-related specimens, whose metabolic activities support the spatial spreading of functional features. Host proteome depicts the major bee physiological activities, supporting the metaproteomics strategy for the simultaneous study of multiple microbial specimens and their host-crosstalks. Altogether, the present study provides a better definition of the structure and function of the bee gut microbiota, highlighting its impact in a variety of strategies aimed at improving/overcoming several current hot topic issues such as antimicrobial resistance, environmental pollution and the promotion of environmental health.

## 1. Introduction

The bee gut is the anatomical section where the major microbial density is hosted and likely the one playing pivotal significance in the One Health perspective [1,2,3]. The gut microbiota of individual bees extends its influence on the entire colony, enabling increased overall productivity [4]. Importantly, bees are most likely involved in the vehiculation of microorganisms. These are, indeed, smaller and lighter than plant seeds, employing bee flies for passively bridging the “extreme” distances existing between the human, animal and environmental fields [5,6]. As such, the gut microbiota composition influences the types of microorganisms transmitted between the environment, flowers, other bees and animals, even at long distances [7]. In this light, understanding the interactions between bees, their gut microbiota members, and the environment is essential for developing strategies to monitor and promote bee health animals’ health and safeguard environmental pollination.

The composition of the bee gut microbiota encompasses primarily bacteria, followed by fungi and other less abundant microorganisms. *Lactobacillus* spp., *Gilliamella* spp., *Bifidobacterium* spp., and *Snodgrassella* spp. are accounted as the major portion of the “core” bacterial fraction. Multiple species of the genus Frischella and Bartonella are considered “noncore” members, acknowledging the dynamic nature observed in diverse environments. Similarly, heterogeneity in the genera Commensalibacter, Bombella, and Apibacter is recorded, although rather commonly identified as the gut microbiota members. Fungal specimens include predominantly Cladosporium, Penicillium, and Hanseniaspora, although other fungal specimens are present depending on the surrounding environment [8].

Functions of the bee gut microbiota members are largely assessed via gnotobiotics models and inferred through the study of third insect ‘biology’ [2]. An elegant review provided by Nowak and colleagues defines bee gut microbiota activities as important as the ones of mammals ‘animals’ [9]. Here, the acidic environment promoted by *Lactobacillus* spp. metabolism, along with the host immune system modulation mediated by *Snodgrasella alvi*, protects against pathogens colonization [10]. By producing biofilm, both *S. alvi* and *Lactobacillus* spp. outcompete pathogenic specimens, besides producing antimicrobial compounds that hinder the growth of harmful bacteria and fungi [3]. Besides the strong involvement in digestive functions and host health status promotion, bee gut microbiota is active in the detoxification of environmental pollutants, including the active response against insecticides and antimicrobial agents [9].

Nevertheless, microbiota architecture and functions are highly dynamic, depending on the life cycle stage, age, social role, and seasonality [1]. In addition, changing microbiota composition across taxonomically related specimens suggests an intimate co-evolution of the host and its microbiota, influenced by a variety of factors such as the genetic background and the environmental surroundings where the bee population flies [1].

Despite the pivotal importance of the bee gut microbiota over the diverse fields of life, only a handful of studies are available in the literature dealing with the definition of the bee gut microbiota [1,4,8,9,11,12,13,14,15]. These studies rely on DNA-based approaches that, although of extreme importance, cannot describe the effective functional armory (i.e., metaproteome) and its dynamics occurring among the microbial specimens and their host, which would be the key to improving the bee (gut) role in a One Health perspective. In the present study, we employ a metaproteomic approach to provide the first explorative study of the functional gut microbiota of *Apis mellifera ligustica* and define the structural and biochemical dynamics occurring among the most common inhabitants of the bee gut. Specifically, the study objective spans from the definition of the functional bacterial fraction, in both structural and functional terms, to the fungal and parasite fractions of this complex community (Figure 1). A discussion on the core structure and function of the bee gut microbiota is provided.

## 2. Results

### 2.1. Metaproteomics Dataset Portrays a Heterogeneous Ensemble of Microorganisms

High-throughput tandem MS measurements resulted in the identification of 806 bacterial proteins across the four biological replicates. The full list of the identified bacterial proteins is provided in Appendix A; the molecular weight distribution of the identified proteins and their sorting over the biological replicates is provided in Appendix A, respectively.

Peptide sequences retrieved from the fungal database-dependent search enabled the identification of 889 functional domains of fungal origin, whose distribution across the bee gut microbiota representatives is depicted in Appendix A. Along with bacterial members, the full list of fungal functional repertoire is provided in Appendix A.

The querying of the MS data through a tailored database referring to unicellular parasites returned the identification of 42 proteins, as listed in Appendix A. Of these, 19 entries are quantified and distributed throughout the biological replicates as of Appendix A.

### 2.2. Bacterial Protein Repertoire Elucidates the Complex Structure and Function of the Bacterial Fraction

The identified protein repertoire portrays a complex and heterogeneous bacterial community inhabiting the bee gut. The architecture of the metabolically active bacterial community has been assessed at the family, genus, and species level (Figure 2 and Appendix A, respectively). Bacterial families such as Lactobacillaceae, Bifidobacteriaceae, Bacillaceae, Neisseriaeae, and Microbacteriaceae are the entries recording the highest contribution in terms of overall protein abundance. Other specimens, including Orbaceae, Clostridiaceae and Alcaligenaceae, show good concordance across the biological replicates, although represented by a minor abundance of proteins (Figure 2A). Sample similarity based on the bacterial family composition ranges from 77.33% to 91.65%, as shown in Figure 2B, suggesting that a portion of the bacterial community is strictly host-related.

Nevertheless, a high interindividual variability occurs among the quantitative compositions of the biological replicates. The geometric class plot (Figure 2C) visualizes a variable species abundance distribution pattern for each biological replicates with a changing percentage of species through the abundance classes. Similarly, alpha-diversity evaluation through the most common species richness estimators depicts different species accumulation patterns for each of the bacterial communities represented (Figure 2); interestingly, the bacterial protein dataset results in a variable alpha-diversity by all the queried estimators (i.e., Sobs, Chao1, Chao2, Jacknife1, Jacknife2, Bootstrap, MM, and UGE), indicating that the changing patterns concern bacterial specimens from all class of abundance throughout the samples kept as the representative of the bee gut microbiota (Figure 2D).

The protein inventory portrays the functional concern of the bacterial fraction in the four biological replicates. Sorting the bacterial proteins in the “biological processes” of the Gene Ontology data repository underlines a heterogeneous functional array of the investigated bacterial communities, with an average similarity ranging from 53.59% to 74.71% of the identified biological processes (Appendix A). A functional assessment on a quantitative basis is shown in Figure 3. Here, the cumulative expression of the proteins specifically belonging to the biological processes is visualized on a color key basis, providing evidence of a changing functional profile throughout the samples. Interestingly, most of the “shared” biological processes are featured by diverse color codes indicating a diverse intensity by which the bacterial members are concerned in such functions (Figure 3), likely because of the strict interdependence between the bacterial community and its host. Thus, sorting the bacterial proteins according to various classification criteria by querying a plurality of data repositories represents a suitable strategy for thoroughly assessing the functional profile of the bacterial fraction harboring the bee gut microbiota. The functional categorization of the identified bacterial proteins as of the biological processes and molecular functions of the Gene Ontology data repository is provided in Appendix A. However, a heterogeneous functional profile is observed by all the functional classification criteria employed in our study, suggesting a strong influence of the host on the functional assets of the bacterial members inhabiting the bee gut.

Besides the above interindividual variability, functional investigation of the bee gut microbiota shed light on a variety of biological activities the intestinal bacteria are concerned with, most of which have great potential in the One Health context. Sorting the bacterial as “pathways” of the UniProt data repository depicts a stable bacterial community involved in common anabolic pathways such as Cell wall biogenesis, Nucleotide-sugar biosynthesis, and amino acid biosynthesis, among others (Figure 3E). Nevertheless, a volume of the bacterial protein abundance is devoted to Antibiotic biosynthesis and isoprenoid biosynthesis, suggesting the presence of a microbial armory to maintain the homeostatic balance (Figure 3).

Proteins sorted according to the cellular component of Gene Ontology reveal an important involvement of the bacterial communities in the antimicrobial resistance traits, as pointed out by the identification of bacterial proteins belonging to the “outer-membrane bounded protein”, “extracellular matrix” and “ATP-binding cassette complex”; besides the expression of virulence factors as the “toxin-antitoxin complex”, “bacterial flagelli”, “capsule” and “spore wall”, (Figure 3F).

A deeper characterization of the bacterial functions by digging into the molecular functions of the bacterial proteins confirms the above functional concern. Linking the molecular functions of proteins to the bacterial specimens to which the proteins belong provides a clear snapshot of the ongoing activities and the functional contribution each bacterial member provides (Figure 4, Appendix A). Altogether, members of the family Bacillaceae, Sphingomonadaceae, and Microbacteriaceae are active on a heterogeneous array of molecular functions, with Bacillaceae showing proteins with toxic activity. Members of the family Bacillaceae and Paenibacillaceae are concerned with beta-lactamase activity and penicillin-binding as it is witnessed by the expression of penicillin-binding proteins (PbP), Metallo beta-lactamase and serine hydrolase. Moreover, Nocardiaceae specimens are concerned with rifampicin resistance as a result of the rifampicin monooxygenase expression. Also, members of Bacillaceae, Clostridiaceae, Microbacteriaceae and Sphingomonadaceae are active in the expression of proteins with transmembrane transporter activity and ABC-type transporters such as multidrug ABC transporters and the Cation Diffusion Facilitator (CDF) family proteins. Similarly, Lactobacillaceae bacteria, among others, are concerned with efflux transmembrane transporters, including TolC protein family members and the Resistance-Nodulation-Division (RND) family transporters, whilst Paenibacillaceae, along with members of other families such as Neisseriaceae, Bacillaceae, Microbacteriaceae, and Corynebacteriaceae are highly involved in binding and rearranging their genomic assets, as indicated by their involvement in molecular functions such as “double-stranded DNA binding”, “class-I endonuclease activity” and “ligase activity” (Figure 4, Appendix A).

### 2.3. Functional Concerns of the Fungal Fraction Support Bacterial Findings

Composition of the fungal community, accomplished by retrieving as much as 12,000 peptides into UniPept, results in the identification of a total of 163 fungal families (Appendix A). Figure 5 displays the core composition of the fungal fraction inhabiting the bee gut intestine. The composition of the “core” fungal fraction appears stable and conserved among the biological replicates. Family Aspergillaceae is the most represented, with an average abundance of 11.2% of the fungal protein repertoire. Also, significant representation is devoted to the Glomerellaceae and Nectriaceae families, with an average representation of 3.1% and 3.6%, respectively. Minor changes are observed for the families Cordycipitaceae, Ceratobasidiaceae, Physalacriaceae, and Clavicipitaceae, which are missing in one of the four biological replicates. On the other hand, the fungal families Schizophyllaceae, Serendipitaceae, Pyriculariaceae, and Lyophyllaceae are exclusively identified in one of the representatives (Figure 5). Structural fungal composition “uniqueness” ranges from 32.7% of the unique families to 13.7% of the fungal family’s uniqueness. However, two of the four biological representatives do not harbor exclusive fungal families and show a structural profile comparable to each other.

Functional characterization of the fungal community in terms of biological processes and molecular functions of Gene Ontology depicts a microbial community involved in a wide array of functions, mostly concordant among the fungal bee gut representatives. Central metabolism and ionic exchanges are among the functions the fungal community is primarily concerned with, as it is supported by the massive expression of proteins with ATP-binding activity and metal ion binding (24.4% and 4.9% of the fungal protein, respectively); nevertheless, other molecular functions and biological processes are also massively expressed by the fungal community such as the DNA binding, RNA binding, DNA repair, and ribosome (Appendix A). In accordance with the findings of the bacterial community, penicillin-binding activity is also recorded in the fungal community, with an average concern estimated to be 0.1% of the fungal protein repertoire. In addition, involvement in the ATP-binding cassette and proteins with transmembrane transporter activities are identified in the fungal metaproteome at an average load of 0.2% and 1.7% of the whole fungal proteins, respectively (Appendix A).

### 2.4. The Unicellular Parasites Compose a Minor Fraction Harboured in the Bee Gut

Unicellular parasites are the less represented microbial fraction in the bee gut metaproteome. Nine different parasitic specimens are identified across the biological replicates. Altogether, Carpediemonas and Flabellula specimens are the most represented parasite specimens, with an average representation of 52.2% and 26.6%, respectively, out of the parasite protein repertoire. Although the qualitative composition is conserved among the biological replicates, quantitative variations are observed in the diverse samples. Flabellula is represented by an abundance of variability ranging from 15.5% to 40.3%. Similarly, *Andalucia* spp. members are rendered by a range of 10.7% to 5.7%. Less abundant specimens belong to Paulinella and Cosmarium, whereas *Pyramimonas* spp. and *Vacuolaria* spp. are identified in two of the four biological replicates (Figure 6). Functional classification is performed by sorting the parasitic proteins into functional classifiers of Gene Ontology (Appendix A). This sorting profile mirrors the involvement of the unicellular parasites in “phagocytosis” and “DNA damage checkpoint signaling” biological processes, among others. Besides, protein families involved in protein metabolism and cellular structures are identified as supported by the identification of the “Class-II aminoacyl-tRNA synthetase family”, “Actin family”, and “ParB family”.

### 2.5. Host Proteome Assessment as the Toolbox for the Study of the Physiological Response

The metaproteomics dataset is also employed for the assessment of the host proteome and its potential role in evaluating the interplay occurring among the host physiology and the diverse bacterial members constituting the bee gut microbiota. Searching the raw spectra against the Apis database of UniProt results in the identification of a total of 182 host proteins, the attribution of which is shared between a plurality of species belonging to this genus. Although all investigated, biological replicates show most of the protein abundance as belonging to *Apis mellifera*, an important fraction of the identified proteins is attributed to *Apis cerana cerana*, *Apis cerana*, and *Apis koschevnikovi*. Other species have been identified based on the identified protein ontology to a minor extent (Appendix A). Host functional interplay is investigated by annotating the host genes into KEGG data repositories. The “general metabolism” is the most concerned pathway the host is active in, followed by the “genetic information processing”, “cellular processes”, and “other systemic functions”. Minor differences are highlighted between biological replicates in a quantitative manner, mostly related to the less represented biochemical pathways such as the “exosomes” and the “cytoskeleton proteins”. However “histone” and “proteasome” polygons hold the promise of important One Health relevance (Appendix A).

## 3. Discussion

A steadily growing bench of evidence supports the role of bees in the One Health context. Bees contribute to the pollination of around 70% of the world’s flowering plants, including crops for human food production, with a strong economic impact, among others [16,17,18,19]. Bees structurally modify the surrounding environments by excavating tunnels and constructing nests and hives. These activities facilitate element turnover in soil and promote soil health, besides contributing to the formation of microhabitats (i.e., novel ecological niches) and providing shelter for other (micro)-organisms [20]. Also, bees take part in the human food chain and serve as a food source for many predators, including birds, mammals, reptiles, and other animals [8]. Thus, the presence of bees in an ecosystem is pivotal for maintaining the overall ecological balance [12,20]. Many of the bee’s functions depend on their gut microbiota. It is demonstrated that bee gut microbiota is involved in insect performance [8], resilience to stressors, and quantity and quality of bee-by products [2,8,14,21]. Acknowledging the above, bee gut microbiota hold the potential of a suitable implementation as a panel of biomarkers for effectively monitoring environmental pollution, besides sensing antimicrobial resistance diffusion and pathogenic traits transmission. On the other hand, covering short-to-mid distances by flight, bees can parallelly vehiculate microorganisms between apparently remote habitats, promoting microbial heterogeneity of the microcosms and the movement of the genetic elements these microorganisms bring along.

In light of all the reasons above, a thorough understanding of the bee gut microbiota extent over the various scenarios would provide key knowledge for the tailored management of the bee populations from a variety of perspectives, including the control of antibiotic resistance diffusion and the spread of pathogenic microorganisms and their virulence traits, spatially. Despite its role, only a handful of studies focused on defining such microbial community composition. To the best of our knowledge, this is the very first exploration of the functional microbial community harbored in the gut of Apis mellifera ligustica, meant as the most abundant specimen in the Mediterranean countries since being employed to produce all the “hive products” for human consumption.

### 3.1. Structure of the Bee Gut Microbiota Has the Potential to Vehiculate Microorganisms of One Health Relevance

The composition and functions of the microbial community are likely prone to change depending on a variety of host-dependent-, diet- and environmental-factors. Nevertheless, being a positional, explorative study, we believe that keeping the bee populations heterogeneous as for the above variables (gender, social role, age, etc.) would ensure catching the highest structural and functional diversity attributable to the specimens harboring the bee gut, regardless of any specific treatment or stratification. Although the employed experimental settings enable the provision of a “core” reference microbiota (or metaproteome), care should be taken while digging into the quantitative features of given structural and/or functional traits, as these might be brought into the sample pools only by defined stratifications of our sample population. This leads to a “dilution effect” that cannot be accurately estimated with the present experimental design. In this light, discrepancies observed in the structure and functions of the bacterial composition across the biological replicates might be partially explained by the above sampling peculiarities. Noteworthy, apiaries employed in the present study were kept in a peculiar geographic area so that influences from the surroundings are brought into the metaproteome, participating in the “normal” definition of the *Apis mellifera ligustica* dataset. Also, the timing chosen for the sampling coincides with the bee’s active period, ensuring they visit diverse niches while foraging, thus making our insects suitable models for the investigation of the bridging potential of bees. Altogether, the metaproteomics data portray a heterogeneous ensemble of microbial kingdoms, with the bacterial fraction being the dominant one with respect to the fungal members and the unicellular parasites harbored in the same ecological niche. Here, the structural characterization of the bacterial fraction of the metaproteomics data agrees with previous DNA-based studies. The genus commonly identified in the bee gut as one of the 16S rRNA gene sequencing is also identified in the bacterial protein repertoire as a means of the relative families. The metaproteomics-based assessment of the bacterial fraction at the family level is preferred, although identified protein annotation enables the structural assessment up to species or even strain level. Indeed, the sequence likelihood occurring among taxonomically related specimens might lead to a less accurate description of the bacterial community at the lowest taxonomical levels. This is particularly true in the context of microbial communities composed of multiple and heterogeneous entities, supporting the adoption of a “broader” (i.e., family) taxonomic level while assessing the microbial composition based on metaproteomics data [22]. Minor changes occur in the depiction of bacterial architecture compared with the bacterial community assessment via microbial community profiling techniques. In this view, changes concerning the less abundant specimens are expected because the different investigation approaches target diverse macromolecules (i.e., DNA and proteins). In this light, targeting proteins appeared to resemble the real microbial community condition more accurately than the 16S rRNA gene sequencing is capable of, acknowledging the longer time required for the nucleic acid to score the presence (or activity) of less abundant specimens in spite of changes in the protein abundance and/or transcription profiles [23,24].

Interestingly, the bacterial community composition assessed the presence of commonly known pathogenic families of humans and animals (e.g., Clostridiaceae, Vibrionaceae, and Mycobacteriaceae) besides the identification of Listeriaceae, a foodborne-relevant family. Furthermore, although excluded by the routinary controls employed in beekeeping settings, the metaproteomics data underline the weak presence of Paenibacillaceae. This is the immediate family of *Paenibacillus larvae*, the causal agent of American Foulbrood, one of honeybees’ most devastating infectious diseases [25]. Besides the common inhabitants of the bee gut, this study highlights the identification of bacterial families typically belonging to other spheres of life, such as the environment, animals and humans. 

Members of the family Burkholderiaceae are indeed considered truly environmental saprophytic organisms [26]. This includes phytopathogens, opportunistic pathogens, and primary pathogens for humans and animals, providing a clear example of the potential role of bees in bridging the three major sectors of life [26]. Similarly, the family Sphingomonadaceae includes rod-shaped bacteria commonly found as components of biofilm of diverse environments [27]. The pathogenic potential of these members is negligible; instead, this family shows high potential in the bioremediation of bisphenol A-contaminated areas [28], suggesting how the implementation of the bee population can potentially serve as an active strategy of bioremediation. Similarly, Carnobacteriaceae are ubiquitous lactic acid bacteria commonly isolated from cold and temperate environments [29]. In addition, members of the family Carnobacteriaceae frequently harbor in fish, meat, and dairy products, contributing to food spoilage prevention and the inhibition of pathogenic microorganisms. Its implementation is yet occurring in aquaculture to enhance fish production and quality [29,30]. Other bacterial families relevant to One Health include Bartonellaceae, Nocardiaceae, and Planococcaceae. These are common inhabitants of soil and freshwater samples, besides being commonly identified in the intestines of diverse vertebrates [31,32,33]. From this perspective, the identification of such bacteria in the bee gut provides further confirmation of the bee’s involvement in the taxying of microorganisms that typically belong to diverse and “remote” areas.

Similar to bacteria, the assessment of fungal components of the bee gut microbiota lists commonly identified fungal specimens along with other entities more commonly attributed to foreign sources.

Among these, fungal specimens such as *Fusarium* spp. and *Aspergillus* spp. are typical environmental fungi known for their relevance in the human and animal health fields known for their capability to invade host tissues and the production of mycotoxins [34,35]. Also, fungal specimens such as *Madurella* spp. and *Scedosporum* spp. are relevant for their ability to provoke mycosis at various levels of severity in a broad spectrum of animals, ranging from bovines and horses to dogs and cats [35,36,37,38]. More common is the identification of *Ascosphaera apis*, the causal agent of chalkbrood disease, one of the most significant fungal infections of the honey bees [39], along with the stonebrood disease, which is sustained by *Aspergillus* spp. [40]. Again, routine tests performed in beekeeper settings fail to provide any positivity to these recurrent infections. Nevertheless, the identification of the proteins belonging to these specimens represents a clear indication of their presence in a metabolically active status. Thus, further studies are needed to assess the presence of and extent to which such pathogenic specimens are tolerated in honeybee breeding and evaluate the potential of metaproteomics in the early detection of this devastating disease.

Evaluating the unicellular parasitic fraction in the bee gut contributes to this kingdom relative to the overall microbial community. The reason behind this proportion might be ecological, emphasizing the myriads of relationships occurring with the bacteria and the fungi that share the same ecological niche. Nevertheless, the identification of the unicellular parasites, accomplished based on the metaproteomics data, supports the trend previously seen in fungi and bacteria with the identification of specimens typically attributable to marine and freshwater environments (e.g., Paulinella and Flabellula) [41,42] other than the presence of common intestinal parasites belonging to the Entamoeba genus [43]. Taken together, the parasite data support the pivotal role of the bees in bridging diverse micro-environments, including the movement of their “microbiological signature”. Moreover, a metaproteomic-based assessment of the bee gut microbiota composition underlines the suitability of this approach in mirroring the microbiological fingerprint of the environment, humans and animals, besides confirming the importance of bees as bioindicators of One Health relevance. However, it is not excluded that amending the sample preparation protocols might yield higher metaproteome coverage, especially regarding the fraction of unicellular parasites, for which further tailored studies are desirable.

### 3.2. Metaproteomics Exploration of the Bee Gut Microbiota Underlines Functional Details of One Health Concern

As an explorative study, here we provide, for the first time, complementary knowledge on the functional assets of the diverse microbial specimens inhabiting the bee gut, regardless of experimental variables and/or treatments as commonly employed in metaproteomics. In this view, employing pooled heterogeneous samples ensures a higher coverage of the functional array of bacteria, fungi, and unicellular parasites. This is mirrored by a weak lack of concordance recorded among samples. However, it enables a broad feeding of the functional peculiarities attributable to the bee gut microbiota in standard conditions. As expected, a wide array of bacterial biological processes and molecular functions are attributed to central metabolism and, more specifically, to anabolic reactions. This lays the groundwork for defining the common metabolic routes involved in the bacterial fraction.

Linking bacterial functions with the relative specimens supports the One Health relevance of the obtained outcomes. For instance, the biological process “spore formation” is linked to members of the family Paenibacillaceae. The formation of spores is among the major routes employed by *Paenibacillus larvae* for disseminating the infection across hives [25,44]. Moreover, bacterial concern in other biological processes, such as flagellar production, microbial competition, and toxin–antitoxin systems, indicate a variety of virulence factors that are prone to be vehiculated across the various ecological niches visited by the bees during their flights.

The functional features of the metaproteome reveal a warning about the involvement of the bacterial community in terms of virulence and antimicrobial resistance diffusion, suggesting the applicative potential of bees to sense circulating resistance and virulence traits across environments. Here, identifying penicillin-binding protein (PbP) expression by Bacillaceae and Paenibacillaceae is predictive of such bacterial members’ concern either in the expression of virulence factors or the escape from antibiotic activity (or even both). The expression of PbP in *Bacillus subtilis* and *Paenibacillus larvae* has been linked to sporulation, which is meant to be an effective virulence factor [45]. On the other hand, PbP expression in *Bacillus thuringiensis* has been associated with antibiotic resistance, suggesting its level is a suitable biomarker for sensing beta-lactams [46]. In the present study, the microbiota involvement in beta-lactam resistance is further supported by the expression of proteins belonging to the beta-lactamase superfamily. Accordingly, our data reveal the expression of rifampicin monooxygenase by Nocardiaceae. In *Nocardia farcinica*, rifampicin monooxygenase has been recently described as a flavin-dependent enzyme that catalyzes the rifampicin decomposition by hydroxylation to 2′-N-hydroxy-4-oxo-Rifampicin in the presence of NADPH and oxygen. The latter exhibited two orders of magnitude less activity as both antimicrobial compound and bacteriostatic molecule [47]. In this light, rifampicin resistance has also been observed in other resistome surveys, mainly using Bifidobacterium and Snodgrasella genera, making bees a suitable bioindicator for sensing environmental virulence and resistance traits [48]. 

Also, in a previous study featuring the resistome of two species of bees (*Apis mellifera* and *Apis cerana*) [48], *Gilliamella apicola* was found to be among the major contributors to the resistance against beta-lactams, among others. Accordingly, antimicrobial susceptibility tests performed in a previous study reported that over 50% of the bacterial isolates demonstrated resistance to penicillin, among other antibiotic classes, suggesting the potential role of bees as the bioindicator of the circulating environmental AMR [49]. Moreover, honeybee exposure to veterinary drugs highlights modest changes in the microbiota architecture as compared to the qualitative rearrangements featured by increased AMR genes as detected by qPCR [50]. Interestingly, no oxytetracycline resistance has been observed in the present study, which is somehow common among the beekeepers that constantly administer this antimicrobial compound for the control of larval pathogens [51] yet supporting the accuracy of this model as the effective “biological spy” of the circulating antimicrobial resistance.

Other proteins commonly responsible for the mechanisms of antimicrobial resistance include efflux pumps, e.g., multidrug ABC transporters are common efflux pumps generally responsible for chemoresistance and, more broadly, warrant protection from xenobiotics [52]. Analogously, the TolC protein family identified in our metaproteomic data is putatively involved in the active export of a heterogenous ensemble of molecules, playing a pivotal role in conferring microbial specimens with both virulence and multidrug resistance [53]. Similarly, RND efflux transporters catalyze the active efflux of many antibiotics and chemotherapeutic agents [54], witnessing how multiple microbial specimens accomplish similar biological functions by acting on a diversified biochemical route.

It is worth noting that the metaproteomics dataset underlines the identification of CDF transporters, a ubiquitous family of heavy metal transporters, holding great potential in human health and bioremediation [55], thus providing further support for the value of the bee microbiota for the detection of contaminants as well as their potential implementation for the effective bio-restoration of polluted environments.

Altogether, the production of intermediate and/or other antimicrobic resistance mechanisms may indicate the previous exposure of the insects to antibiotic molecules and pollutants, either directly or indirectly. In addition, this raises the likelihood of spatial diffusion of the antimicrobial resistance traits, acknowledging the role of bees in connecting distant areas. Focusing on Antimicrobial Resistance Genes (ARGs), Sun and colleagues proved the high potential of the bee gut microbiota members in the transfer of AMR traits among microbial members and across environments, keeping the AMR dissemination cycle active and thus supporting the dynamic nature of the microbiota depending on the host genotype and the surrounding environment [48]. Accordingly, similar outcomes are observed in the functional detailing of the fungal community, supporting the above findings by identifying common functional patterns in two diverse microbial kingdoms. Nonetheless, viewing the bee gut as the “container” of a dense and heterogeneous ensemble of microorganisms, it is plausible to consider the risk of intra- and inter-domain movement of antimicrobial resistance traits, both genetically and non-genetically determined [4,7,56]. Here, further tailored studies urge to be performed to better understand the dynamics in the spread of the resistance traits, especially in light of the important involvement of all the surveyed microbial domains (i.e., bacteria, fungi, and parasites) in rearranging their genetic makeup. Moreover, acknowledging the quick and efficient communication occurring among microbial specimens, even taxonomically unrelated, further investigation on the mechanisms regulating the nongenetically determined resistance to antimicrobial compounds would be desirable in this sample matrices due to the rather predictive of the microbiological milieu in a variety of different ecological niches.

## 4. Materials and Methods

### 4.1. Bee Population and Sample Collection

The bee population, heterogenous in terms of gender, age and social role, was kindly provided by a local beekeeper in the province of Catanzaro, Italy. Apiaries were collected at the University Campus (38°52′05.3″ N 16°34′55.8″ E), in a strip of land characterized by a rather heterogeneous environment in a narrow space; this includes sea hills, lakes, and streams. Also, employed bees arise from a wider project aimed at biosensing purposes, and no medicaments were administered within six months before experimental enrollment. Samples were collected during insect activity (i.e., from May until July). After being caught, insects were transported to the university lab, where bees were kept at −20 °C for 15 min and sacrificed by saturating the environment with CO_2_. The bees’ guts were obtained by pulling them out from the sting using sterile forceps. Extracted guts were stored in pre-chilled tubes, randomly pooled into four independent aliquots of 0.5 g each (corresponding to an average of 30 insects per sample), as biological replicates, and stored at −80 °C until further analysis.

### 4.2. Microbial Fraction Isolation and Metaproteome Extraction and Digestion

Microbial fractions were extracted by multiple steps of centrifuge/resuspension as detailed in a previous protocol [24]. Then, the recovered microbial cells were subjected to the extraction of the metaproteomes and in-solution trypsin (Promega, MA, USA) digestion as per previously employed procedures [57].

The recovered peptides were purified and desalted using Zip-Tip C18 tips (Millipore, Billerica, MA, USA) and dried in a SpeedVac (Eppendorf, Milan, Italy) until the LC-MS/MS measurements.

### 4.3. LC-MS/MS Measurements

The dried peptide mixture was resuspended in 0.1% formic acid and subjected to UPLC-MS/MS measurement by Dionex UltiMate 3000 RSLC nano system (Thermo Scientific, Sunnyvale, CA, USA) and Orbitrap Fusion Lumos nanoESI-MS/MS (Thermo Fisher Scientific, Waltham, MA, USA), respectively. The employed instrumental settings and measurement protocols are those previously validated by Marini et al. [58].

The mass spectrometry proteomics data were deposited to the ProteomeXchange Consortium via the PRIDE [59] partner repository with the dataset identifier PXD043896.

### 4.4. Bioinformatic Data Analysis

Tandem mass spectrometry raw data were processed through MaxQuant (v. 2.3.1.0, Max Plank Institute of Biochemistry, Martinsried, Germany) set on LFQ modality and searched against in-house databases. Independent searches were performed against a database comprising the most identified bacterial families in the literature [8]. Other database-dependent searches were performed against fungi (UniProt “Fungi”, taxID:4751), unicellular parasites (UniProt “Protist”, key search term), and the host-tailored database (UniProt “Apis”, taxID:7459). Peptide identification and protein inference were accomplished by setting cysteine carbamidomethylation as the fixed modification and methionine oxidation as the variable modification. Two missed cleavage sites were allowed for the in silico protease digestion, and peptides had to be fully tryptic. All other software parameters were set as a default, including a peptide and protein FDR < 1%, at least one peptide per protein, a precursor mass tolerance of 4.5 ppm after mass recalibration, and a fragment ion mass tolerance of 20 ppm. Taxonomic information was inferred according to the protein description obtained from the UniProt database annotation (https://www.uniprot.org/, accessed on 15 December 2023). Identified proteins were functionally classified into COG “biological processes”, “molecular functions”, and “cellular component” functional categories through the direct cross-linking enabled by the UniProt data repository.

Comparisons between samples were performed for each dataset and presented as Venn diagrams using the Venny online tool (accessed on 10 January 2024). Protein abundance indexes of the identified proteins (LFQ values) were subjected to statistical investigation via Primer7 v.7 statistical software (v7, PRIMER-E, Plymouth, UK) [60]. Principal coordinate analysis (PCoA) was calculated on the Bray–Curtis dissimilarity matrix, which, in turn, was calculated on the square root transform of the protein LFQs. Statistical differences across sample composition and functional signatures were calculated by performing dedicated statistical surveys. Alpha diversity indexes were computed as Simpson, Chao1 Chao2, Jacknife1 Jacknife2, MM, and UG indexes. Similarity percentage analysis (SIMPER) was also performed to select the features driving dissimilarities between the biological replicates. Heat maps visualizing microbial community composition and the functional classification of the identified proteins were drawn using heatmap.2 provided by the gplots package implemented in R v.3.1.2 software (v. 3.1.2, R Foundation for Statistical Computing, http://www.R-project.org, accessed on 20 January 2024).

## 5. Conclusions

Bees are crucial for maintaining healthy ecosystems, promoting plant reproduction, supporting biodiversity, and ensuring the availability of food resources for numerous organisms. Their presence and activities have far-reaching effects on the stability and functioning of natural ecosystems. Bee roles are to be attributed to the “superorganism”, recognizing the pivotal role of the associated microbiota. This is, indeed, often forgotten and only rarely investigated. Here, the microbial community inhabiting the bee intestine was split into three macro areas: bacteria, fungi, and unicellular parasites. Each area is characterized in a structural and, for the first time, in a functional manner by means of the metaproteomics approach, providing a knowledge base of the structure and functions plausibly present in the gut microbiota of the bee *Apis mellifera ligustica*. The results obtained are suggestive of the One Health relevance of the bees through a side-way of their activity most recognized and acknowledged. We are confident that improving our knowledge of the potential of bees throughout their flight can be exploited to solve, at least in part, some of the hot topic issues of our era, including the assessment/monitoring of antimicrobial resistance, evaluation of environmental pollution, and the promotion of environmental health, in line with the One Health concept.

## Figures and Tables

**Figure 1 ijms-25-03739-f001:**
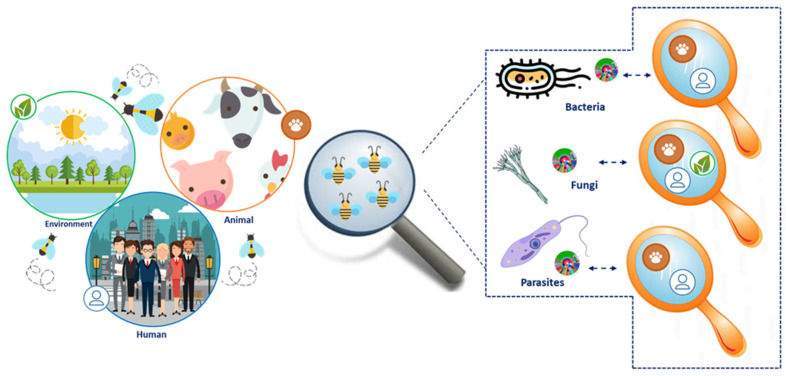
Experimental design overview. The figure summarizes the experimental design employed in the present study along with the major outcome registered for the bacterial, fungal, and unicellular parasites fraction.

**Figure 2 ijms-25-03739-f002:**
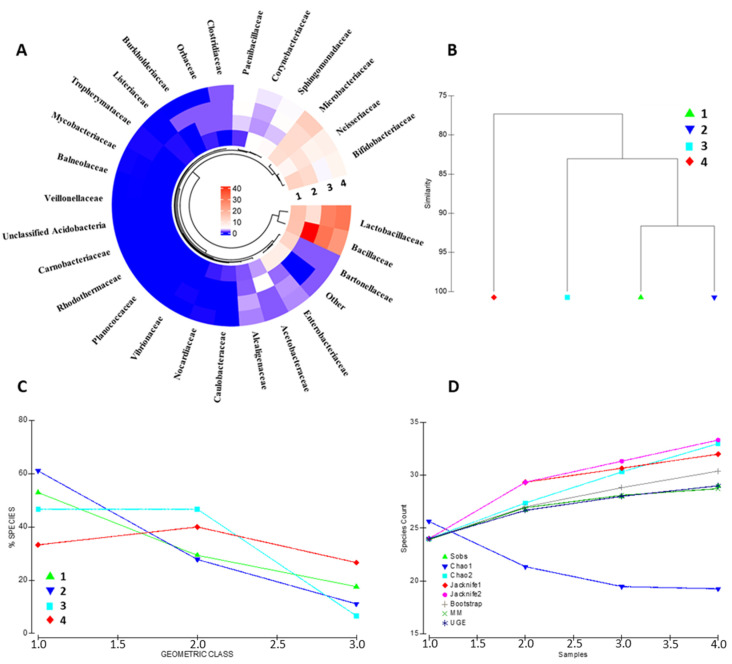
Bacterial community composition assessment. The figure summarizes the most relevant bacterial families identified in the four biological replicates; color code is relative to the cumulative abundance of the proteins attributable to the identified bacterial families (**A**). Panel (**B**) depicts the similarity plot, computed on the bacterial composition data, of the four biological representatives. The geometric class plot of Panel (**C**) visualizes the bacterial families of each of the biological replicates as clustered on the basis of pre-determined abundance ranks. Panel (**D**) shows the most employed indexes for the assessment of the alpha diversity in each of the samples considered in the present study.

**Figure 3 ijms-25-03739-f003:**
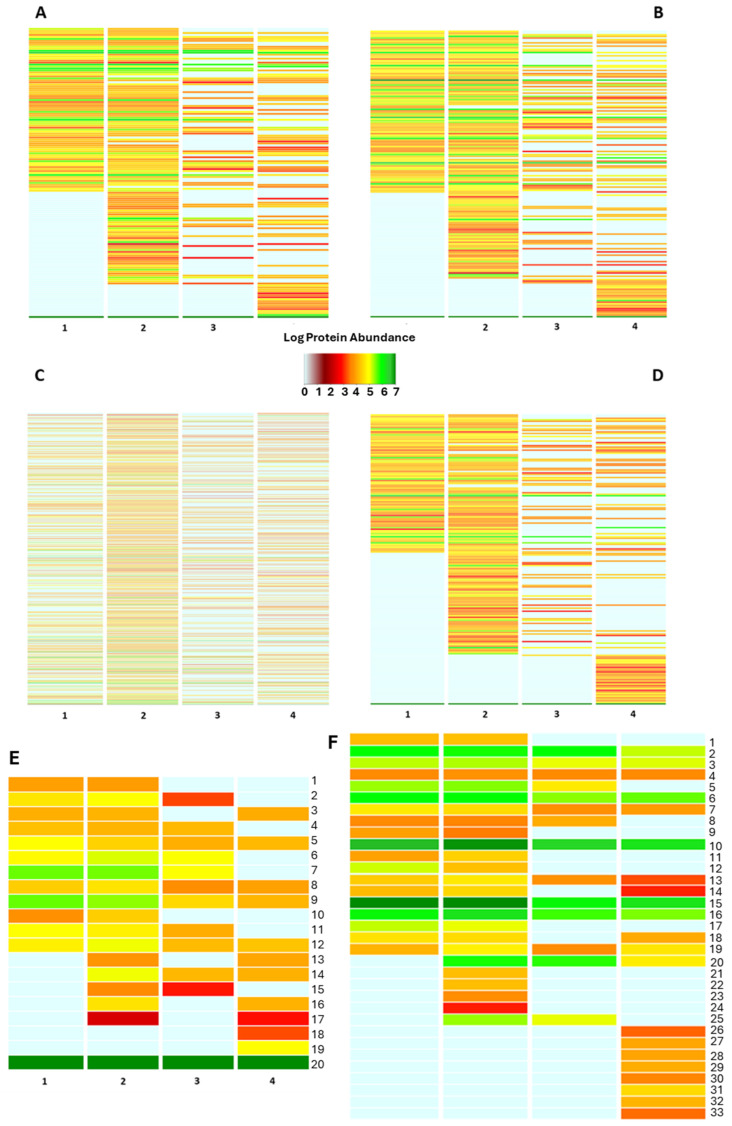
Functional classification of the bacterial protein repertoire. The figure summarizes the functional pattern (s) of the bacterial community harboring the bee gut as assessed by sorting the bacterial protein abundances into biological processes (**A**), molecular function (**B**), PFAM domains (**C**), protein families (**D**), pathway (**E**) and cellular component categories of Gene Ontology data repository. Pathway functional classes: 1_Pyrimidine metabolism; 2_Purine metabolism; 3_Protein modification; 4_Porphyrin-containing compound metabolism; 5_Nucleotide-sugar biosynthesis; 6_Metabolic in-termediate biosynthesis; 7_Glycolipid biosynthesis; 8_Cofactor biosynthesis; 9_Cell wall biogenesis; 10_Carbohydrate metabolism; 11_Amino-acid degradation; 12_Amino-acid biosynthesis; 13_Sulfur metabolism; 14_Quinol/quinone metabolism; 15_Lipid metabolism; 16_Isoprenoid biosynthesis; 17_Antibiotic biosynthesis; 18_Phospholipid metabolism; 19_Glycan biosynthesis; 20_Unknown. The annotation for each of the above panels is also provided in Appendix A. (**F**) Bacterial protein sorting into the “cellular component” categories of Gene Ontology data repository: 1_Toxin-Antitoxin complex; 2_Ribonucleoprotein complex; 3_Catalytic Core F(1); 4_Proteasome complex; 5_Pore complex; 6_Plasma membrane; 7_Periplasmic space; 8_Oxoglut. Dehydrog. Complex; 9_Outer-membrane bounded; 10_Membrane; 11_Intracellular membrane bounded org.; 12_ Host cell cytoplasm; 13_Extracellular region; 14_DNA-directed RNA polym. Complex; 15_Cytoplasm; 16_ Cell outer membrane; 17_Capsule; 18_Bacterial-type flagellum; 19_ATP-binding cassette complex; 20_Extracellular matrix; 21_Cytoplasmic vesicle; 22_Cytochrome complex; 23_Chromosome; 24_beta-galactosidase complex; 25_Bacterial type flagellar basal body; 26_Virus tail; 27_Sulfite reductase complex; 28_Spore wall; 29_Ribosome; 30_Glycolate oxidase complex; 31_Cytosol; 32_Bac. Type flag. Basal body; 33_3-isoprop. Dehydratase complex.

**Figure 4 ijms-25-03739-f004:**
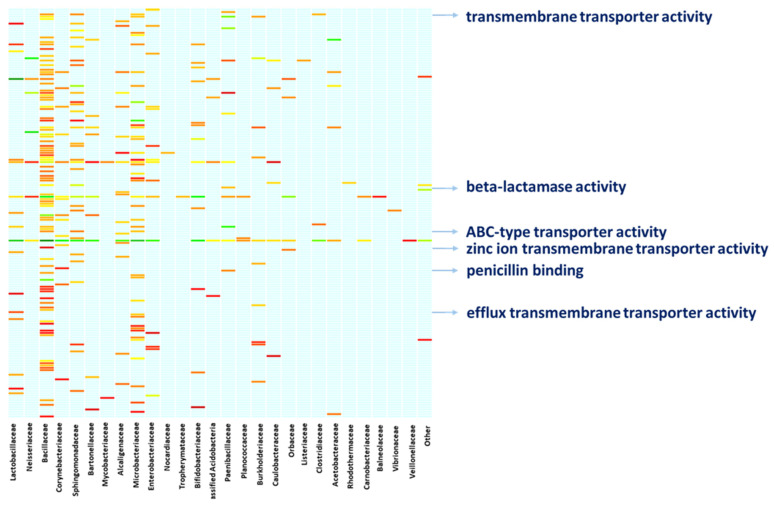
Functional involvement of bacterial families. The figure resumes the functional concern of the bacterial families hosted in the bee gut. The arrows highlight examples of the One Health relevant functions performed by the bacterial fraction, supporting the role bees might cover while bridging the human, animal, and environmental spheres. A full list of the functional entries is provided in Appendix A.

**Figure 5 ijms-25-03739-f005:**
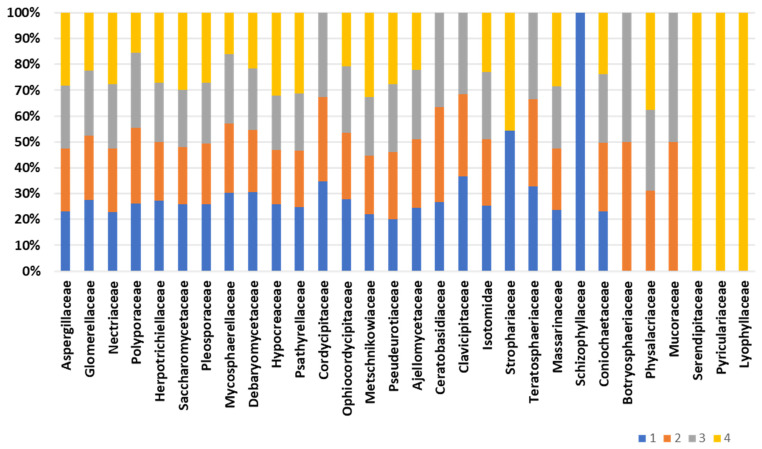
Fungal composition of the bee gut microbiota. The bar chart depicts the composition of fungal families across biological replicates.

**Figure 6 ijms-25-03739-f006:**
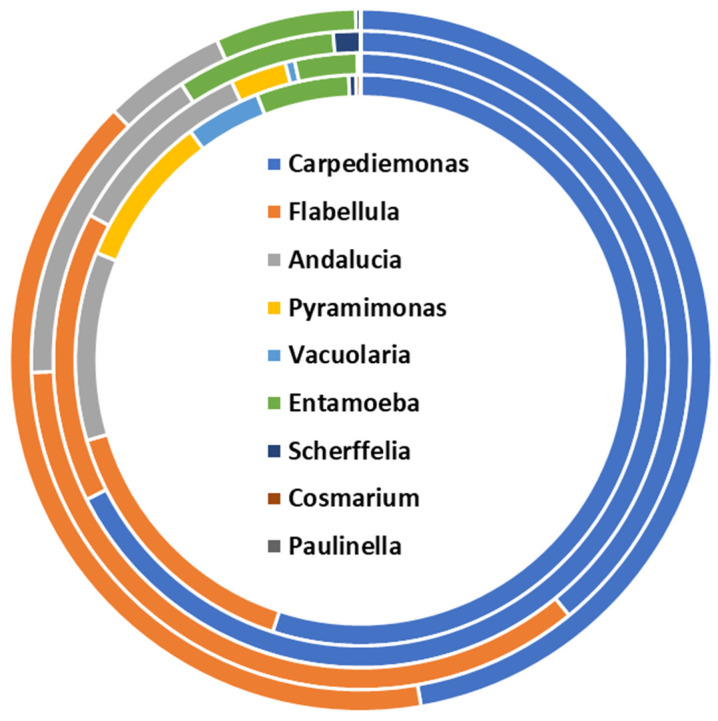
Structural composition of the unicellular parasite fraction of the bee gut microbiota assessed by metaproteomics.

## Data Availability

The data presented in this study are openly available in the PRIDE repository at [59], reference number PXD043896.

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
