# Peer review of "The Bee Gut Microbiota: Bridging Infective Agents Potential in the One Health Context"

_ijms, 2024, doi:10.3390/ijms25073739_

Round 1

Reviewer 1 Report

Comments and Suggestions for Authors

Thank you for this work. The work has clearly addressed the importance of bees in the ecological system. Moreover, the work highlighted the significance of microbiota in the connections between humans, animals and plants. 

I 'd like to see a paragraph about the applied side of this work as some proteins that represent antibiotics resistant genes have a crucial issue.

Introduction is fine and can be strengthened by some important proteins of microbial origin in bee environment.

Methodology: we need some information about location of the apiary. Was the sampling at certain season of the year or it was done many times during the year?

was there any medication given to the bees during the time of collection?

How many bees were used per sample?

Over all the manuscript is suitable for publication in IJMS.

Reviewer 2 Report

Comments and Suggestions for Authors

This ms describes the results of applying metaproteomic techniques to the study of the bee gut microbiota, including bacteria, fungi and parasites (protozoa). The topic is of interest due to current interest in the decline of bee communities and the suggested approach requires intensive work. At the down site, I find the ms too generalist lacking specific examples and/or mechanisms of response that could be extracted from the authors data. As it is the ms results a bit repetitive and I think that it could be shorted. As the general text is reduced, the space might be filled with details and specific examples or mechanisms that examplify major results extracted from the data. I understand this could not be so easy but it would greatly increase the impact, relevance and contribution to the field of the authors results resulting in a potentially outstanding paper instead of the current version which could represent an average publication. The authors have in their hands the possibility to produce a really interesting article.

Some additional comments to be considered:

Please, replace "over the space" by spatial spread/ing, for example in lines 26 and 275, perhaps, also in some other lines.

The use of supplementary material is necessary but I would recommend to attempt to include enough information in the ms so that the supplemented material is only required to gain more details and specific information. An example is Figure 3 which result a bit poor as it is. Perhaps you can highlight some specific functions of interest that are to be commented in the text.

Bees are, generally (except during swarming), relatively local travelers. How does this fit with the hypothesis of vehiculation of pathogens and antimicrobial resistance?

By studying the whole diversity (mostly bacterial and fungi) one might get into additional complexity (due to inter-individual differentiation, diversity, perhaps due to differences in sex, age or status). Have the authors considered to study the "core" among the results for all samples? Perhaps, more simple and effective conclusions could be extracted although this is, of course, just a suggestion.

Identification of proteins as in a metaproteome makes difficult or impossible the identification of novel (to be detected or classified) taxa. This is unlike DNA sequencing analysis (metagenomics). Is there a large or insignificant potential for this potential artifact in the bee guts? Or in other words, could this result in a simplification of results or how could be affecting data?

The authors mention the vehiculation of antibitic resistance by bees after being in contact with antibiotics. This could be clarified because those antibiotics could be from human activity origins or by naturally producing antimicrobials by natural microorganisms. It is known that numeorus microorganisms can produce antimicrobials of different types, for instance, to outcompete others. If this could be like this, the contact with antimicrobials and the mechanisms of resistance could be natural and not human-related. Just a consideration for the authors.

I miss some background on what is known o have been published on bee microbiota including references on this. In parallel, some more general references could be replaced by others more specific on describing detailed information representing the state-of-the-art on bee microbiota. I recommend to be more specific in all aspects of the ms avoiding to fall in too many general comments which impoverish the article.

Round 2

Reviewer 2 Report

Comments and Suggestions for Authors

I consider the ms has improved following the suggestions and could be accepted for publication